# Predicting the Order of Upcoming Tokens Improves Language Modeling

**Zayd M. K. Zuhri**[1]   **Erland Hilman Fuadi**[1]   **Alham Fikri Aji**[1]

## Abstract

Multi-token prediction (MTP) has been proposed as an auxiliary objective to improve next-token prediction (NTP) in language model training but shows inconsistent improvements, under-performing in standard NLP benchmarks. We found MTP's exact future token prediction to be too difficult as an auxiliary loss. Instead, we propose token order prediction (TOP), which trains models to order upcoming tokens by their proximity using a learning-to-rank loss. TOP requires only a single additional unembedding layer compared to MTP's multiple transformer layers. We pretrain models of 340M, 1.8B, and 7B parameters using NTP, MTP, DeepSeek MTP (DS-MTP) and TOP objectives. The results of nine standard NLP benchmarks show that TOP overall outperforms NTP, MTP, and DS-MTP even at scale. TOP models with continued training on math and code also perform better on 4 relevant benchmarks. On the synthetic star graph task, TOP enables pathfinding on graphs where NTP, MTP, and DS-MTP fail. Our implementation and training code is available at https://github.com/zaydzuhri/token-order-prediction.

## 1. Introduction

Current large language models (LLMs) are trained to predict the next token in a sequence during training, an unsupervised learning task referred to as next-token prediction (NTP) (Shannon, 1948; 1951). Although simple, NTP has been very successful in creating powerful language models that can solve complex tasks and even reason over their context.

[1]Mohamed Bin Zayed University of Artificial Intelligence, Abu Dhabi, United Arab Emirates. Correspondence to: Zayd M. K. Zuhri <zayd.zuhri@mbzuai.ac.ae>.

*Proceedings of the 43rd International Conference on Machine Learning*, Seoul, South Korea. PMLR 306, 2026. Copyright 2026 by the author(s).

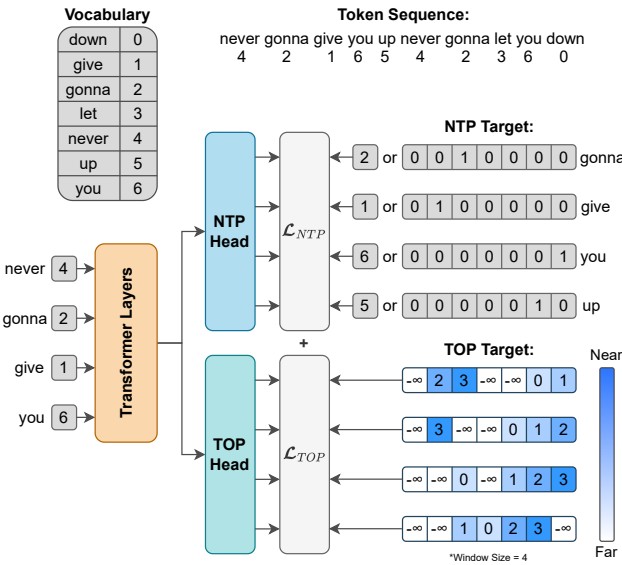

*Figure 1.* An overview of token order prediction (TOP). Given an input token sequence, a vocabulary, a sequence length of 4 and window size of 4, a TOP target sequence is constructed via Algorithm 1. The output hidden representation of the final layer goes to two separate unembedding heads for NTP and TOP. The final loss to optimize is a sum of the NTP and TOP loss.

However, NTP has received various criticisms in recent years. A notable argument by LeCun (2024) claims that NTP at inference time accumulates errors over every time step and inevitably falls off greatly in accuracy. This was however refuted by Bachmann & Nagarajan (2024), in which they argue that the main issue of NTP lies not in inference time error accumulation; rather, that teacher forcing is unable to learn an accurate next-token predictor in the first place. This has motivated the work on alternative or auxiliary LLM training objectives.

Building off ideas such as ProphetNet (Qi et al., 2020), multi-token prediction (MTP) (Gloeckle et al., 2024) has emerged as a relatively successful auxiliary learning task to improve NTP in LLM training. For example. a variant of this method was used in DeepSeek V3 (DeepSeek-AI et al., 2024). MTP adds multiple heads to the end of a transformer (Bahdanau et al., 2014; Vaswani et al., 2017) that each predict a different offset of tokens ahead. All MTP heads share the same trunk of transformer layers, with the

hope that having these auxiliary heads leads to the model learning better internal representations that are considerate of not only the next immediate token, but also future tokens that may come after it. It has been shown that MTP improves performance of LLMs on generative tasks that require lookahead, such as coding and math.

However, MTP shows inconsistent results in general NLP tasks, underperforming in standard downstream benchmarks (Gloeckle et al., 2024, Appendix G). We aim to improve upon MTP by introducing a different auxiliary training objective with the same goal as MTP; enhancing language modeling performance by learning to predict beyond the next token. However, instead of exactly predicting multiple future tokens, we propose that a better training objective is to predict the order of upcoming tokens in the sequence with a learning-to-rank loss. In this paper, we contribute the following:

1. We introduce token order prediction (TOP), a novel auxiliary training loss in addition to NTP to improve language modeling in general.

2. For each of the four training strategies NTP, MTP, DeepSeek MTP (DS-MTP), and TOP, we pretrain language models with sizes of 340M, 1.8B, and 7B parameters on up to 104B tokens.

3. We evaluate these models on standard NLP benchmarks and show that TOP improves on NTP, MTP, and DS-MTP even at scale. We continue training on math and code, observing gains from TOP on generative benchmarks as well. Additionally, the synthetic star graph pathfinding task shows TOP can solve lookahead problems that NTP, MTP, and DS-MTP cannot.

## 2. Background

Next-token prediction (NTP) is the standard training objective for present-day language models. This task is learned by optimizing the cross-entropy loss over the sequence length. Given sequence length $T$, model dimension $D$, vocabulary size $V$ and $\mathbf{x} = \{x_0, \ldots, x_{T+1} \mid x_i \in \mathbb{Z}\}$ as the input token sequence, this loss is written as

$$\mathcal{L}_{\text{NTP}} = -\sum_{t=0}^{T} \log(\mathrm{P}_\theta(x_{t+1}|x_{0:t})) \tag{1}$$

where $\mathrm{P}_\theta$ is the output probability given by the language model with parameters $\theta$. The probability of the next token $x_{t+1}$ given this model is written as

$$\mathrm{P}_\theta(x_{t+1}|x_{0:t}) = \mathrm{softmax}(\mathrm{U}_{\text{NTP}}(\mathbf{h}_t^L))[x_{t+1}] \tag{2}$$

where the hidden representation $\mathbf{h}_t^L \in \mathbb{R}^D$ is generated by a transformer up to the final layer $L$ conditioned on $x_{0:t}$, and

the NTP head $\mathrm{U}_{\text{NTP}} : \mathbb{R}^D \to \mathbb{R}^V$ is a linear unembedding layer to project $\mathbf{h}_t^L$ onto the vocabulary. The probability is taken at the index of the target token $[x_{t+1}]$.

Multi-token prediction (MTP) (Gloeckle et al., 2024) was proposed as an architectural modification that adds additional MTP heads[1] in the form of parallel, singular transformer layers that each output a future token prediction at offset positions. Given $N$ as the number of future tokens to predict (including the next token), the MTP loss can be written as

$$\mathcal{L}_{\text{MTP}} = -\sum_{t=0}^{T} \log(\mathrm{P}_\theta(x_{t+1:t+N}|x_{0:t}))$$

$$= -\sum_{t=0}^{T}\sum_{n=1}^{N} \log(\mathrm{P}_\theta(x_{t+n}|x_{0:t})) \tag{3}$$

If we define $\mathbf{h}_t^{L-1}$ as the hidden representation before the last transformer layer and have $\mathrm{F}_i$ for $i = 1, .., N$ as the MTP heads in the form of singular transformer layers for each future token, and all heads share the same unembedding layer or NTP head $\mathrm{U}_{\text{NTP}}$, then:

$$\mathrm{P}_\theta(x_{t+n}|x_{0:t}) = \mathrm{softmax}(\mathrm{U}_{\text{NTP}}(\mathrm{F}_n(\mathbf{h}_t^{L-1})))[x_{t+n}] \tag{4}$$

MTP promises better performance on generative tasks such as coding, math, and summarization that benefit from the lookahead nature of MTP. MTP also allows the model to do a form of self-speculative decoding, which speeds up inference to some degree. However, MTP does not seem to improve overall language modeling performance on downstream tasks other than those mentioned above, struggling on standard NLP benchmarks, as shown in Appendix G of their paper. Even in generative tasks, MTP harms performance in smaller models and only starts to gain advantage over NTP for models with more than 1 or 3 billion parameters. Furthermore, the paper shows that MTP cannot arbitrarily use a large number of future tokens, where it is shown that 4 future tokens performs better than 8 in coding.

Other MTP variants such as the one used for DeepSeek V3 training have also been used (DeepSeek-AI et al., 2024). We refer to this variant as DS-MTP. Unlike the original MTP, DS-MTP arranges the MTP heads sequentially. Each head after the first one receives concatenated, normalized embeddings from the original tokens up to that offset.

$$h_t^{L-1+n} = \begin{cases} \mathrm{F}_n(\mathbf{h}_t^{L-1}) & n = 1 \\ \mathrm{F}_n([\bar{\mathbf{h}}_t^{L+n-2}; \bar{\mathbf{e}}_{t+n-1}]) & 2 \leq n \leq N \end{cases} \tag{5}$$

$$\mathbf{z}_{t,n} = \mathrm{U}_n(\mathbf{h}_t^{L+n-1}) \tag{6}$$

---

[1] Disambiguation: Architecturally, MTP heads are transformer blocks, as we follow the terminology from Gloeckle et al. (2024). Meanwhile, NTP head and TOP head are linear unembedding layers.

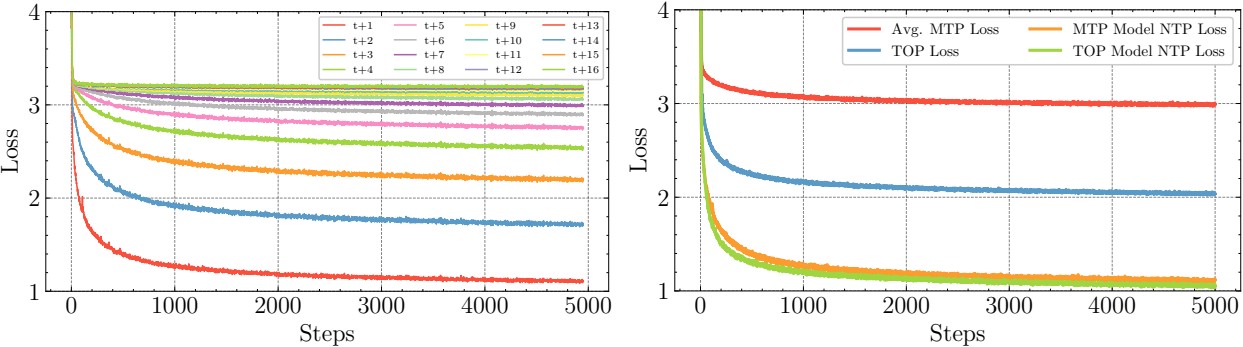

*Figure 2.* Left: Training loss of a MTP transformer with 16 MTP heads predicting tokens at $t+1, ..., t+16$ offsets. Right: Training loss of the MTP model averaged over all 16 heads, compared to the training loss of a same-sized TOP model with window size 16, in addition to the isolated NTP loss of both models.

$$P_\theta(x_{t+n} \mid x_{\leq t+n-1}) = \text{softmax}(\mathbf{z}_{t,n})[x_{t+n}] \quad (7)$$

$$\mathcal{L}_{\text{DS-MTP}} = \sum_{t=0}^{T} \sum_{n=1}^{N} \log(P_\theta(x_{t+n}|x_{0:t})) \quad (8)$$

Where $\bar{\mathbf{h}} = \text{RMSNorm}(\mathbf{h})$ and $\bar{\mathbf{e}}_t = \text{RMSNorm}(E(x_{0:t}))$ denote the normalized hidden states and embeddings, respectively. The unembedding layer $E : \mathbb{Z} \to \mathbb{R}^D$ is shared between MTP heads and the main transformer trunk. Notably, DeepSeek V3 only used $N = 3$ for their MTP training, which means the model only learns to predict the next three tokens. This might indicate a key point of our paper, which is that the objective of MTP is too difficult to be an effective auxiliary loss to NTP, especially for larger lookahead values $N$.

## 3. Motivation

Our hypothesis for why MTP only partially improves language modeling is that MTP is too difficult as a learning objective. If we look at the original MTP paper (Gloeckle et al., 2024), there are two empirical results supporting this argument. First, MTP does not improve the performance of small language models on generative tasks, such as coding. This suggests that a certain capability threshold is required for MTP's multi-token modeling to be effective, which they observe to be in the 1B-3B parameter range. Second, increasing the number of future tokens in MTP does not guarantee a better overall performance. The ideal number of future tokens varies across different tasks. Not only does this make it difficult to determine the maximum number of heads beforehand, it also indicates that there are thresholds of lookahead distance where the difficulty of prediction starts to hurt learning instead of helping it.

To illustrate our argument, we train a small 16M parameter transformer with 16 MTP heads and visualize the training loss of each MTP head in Figure 2. A clear pattern emerges where the losses of predicting the tokens at posi-

tions $t+1, ..., t+16$ arrange themselves from bottom to top. Each future token farther away significantly worsens in loss compared to the immediate next token loss and shows a decreased rate of loss descent, indicating the difficulty of exactly predicting far ahead. We believe that relaxing this MTP objective will make it more useful as an auxiliary loss. Compared to a similarly sized model with the TOP objective at window size 16, we see that the TOP loss is lower. Additionally, the final isolated NTP loss of the TOP model is also lower than the MTP model.

## 4. Method

### 4.1. Token Order Prediction

We propose token order prediction (TOP), a novel auxiliary training loss for language modeling. Given a token sequence $\mathbf{x} = \{x_0, \ldots, x_T \mid x_i \in \mathbb{Z}\}$, we construct a TOP target sequence $\mathbf{y} = \{y_0, \ldots, y_T \mid y_i \in \mathbb{Z}^V\}$, where $V$ is the vocabulary size. Each entry $y_t$ assigns a proximity score to every token in the vocabulary. The scores rank tokens in descending order by how soon they first appear after $x_t$ in $\mathbf{x}$, so tokens that appear sooner receive higher scores. We also introduce a hyperparameter, window size $W$, within which the token order is evaluated. This target sequence can be constructed via Algorithm 1.

Algorithm 1 constructs the TOP targets by scanning the sequence once in reverse while tracking, for every vocabulary token, its next occurrence position. It initializes the target tensor $\mathbf{y} \in \mathbb{R}^{T \times V}$ to $-\infty$ so tokens that never appear in the future context receive no score, and sets an auxiliary array $\mathbf{n}[v] = T + W$ for all $v$ as a sentinel "not seen yet" next-occurrence index. Iterating $t$ from $T + W - 1$ to 0, the algorithm first updates $\mathbf{n}[\mathbf{x}[t]] \leftarrow t$ (for valid tokens in the vocabulary i.e. if it is not the padding token and its index lies in [0, V-1]), so that $\mathbf{n}[v]$ always points to the future position closest to where the token $v$ appears, relative to the current $t$. For output positions $t < T$, it then computes for each token

**Algorithm 1** Convert a token sequence to a TOP target sequence

---

**Goal:** For each position, compute a proximity score to the next occurrence of every token within a window of size $W$.

**Require:** Token sequence $\mathbf{x}$ of length $T + W$, vocab size $V$, window size $W$.

**Ensure:** Tensor $\mathbf{y}$ of shape $(T, V)$

1: Initialize $\mathbf{y} \leftarrow -\infty$
2: Initialize $\mathbf{n}[v] \leftarrow T + W$ for all $v \in [0, V-1]$
3: **for** $t \leftarrow T + W - 1$ down to 0 **do**
4:    **if** $\mathbf{x}[t] \in [0, V-1]$ **then**
5:       $\mathbf{n}[\mathbf{x}[t]] \leftarrow t$
6:    **end if**
7:    **if** $t < T$ **then**
8:       **for** each $v \in [0, V-1]$ **do**
9:          $d \leftarrow \mathbf{n}[v] - t$
10:          **if** $0 < d \leq W$ **then**
11:             $\mathbf{y}[t, v] \leftarrow W - d$
12:          **end if**
13:       **end for**
14:    **end if**
15: **end for**

---

$v$ the distance $d = \mathbf{n}[v] - t$ to its next occurrence and, if $0 < d \leq W$, assigns the proximity score $\mathbf{y}[t, v] = W - d$, giving higher values to tokens that appear earlier and leaving all others at $-\infty$. To better understand this target sequence, please refer to the visualization in Figure 1. In practice, we have an optimized Triton kernel for this function that creates the target sequence on the fly during training and practically incurs no overhead. Alternatively, one could also pre-process an entire dataset beforehand.

To train the model to order upcoming tokens as in the target sequence, we borrow a loss formulation from the learning-to-rank literature (Pobrotyn et al., 2020), more specifically from ListNet (Cao et al., 2007) called listwise ranking loss. For an input with a list of documents, we can define listwise ranking loss as:

$$\mathcal{L}_{\text{ListNet}} = -\sum_{t=0}^{T} P(y_t) \log P(\widehat{y_t}) \qquad (9)$$

Where $P(y_t)$ is the top-one probability of the true labels and $P(\widehat{y_t})$ is the top-one probability of the predicted scores. For TOP, we will use softmax to get the top-one probability and cross-entropy for the distance metric. Hence, the TOP loss formula becomes:

$$\mathcal{L}_{\text{TOP}} = -\sum_{t=0}^{T} \text{softmax}(y_t) \cdot \log(\text{softmax}(\mathrm{U}_{\text{TOP}}(\mathbf{h}_t^L)))$$
$$(10)$$

*Table 1.* Complexity analysis of different multi-token prediction methods for training. $D$ is the hidden size, $V$ is the vocabulary size, and $N$ is the number of future tokens.

| METHOD | FLOPs | PARAMETERS |
|---|---|---|
| MTP | $(N-1)(24D^2 + 2DV)$ | $(N-1)(16D^2 + 2D)$ |
| DS-MTP | $(N-1)(30D^2 + 2DV)$ | $(N-1)(16D^2 + 2D)$ |
| TOP | $2DV$ | $DV$ |

Note that $\text{softmax}(\mathrm{U}_{\text{TOP}}(\mathbf{h}_t^L))$ is not a probability distribution of only the next token, hence we do not write it as $P_\theta$. The correct way to think about it is to view $\mathrm{U}_{\text{TOP}}(\mathbf{h}_t^L)$ as the model prediction of the ranking in the form of proximity scores, and $\text{softmax}(\mathbf{y}) \cdot \log(\text{softmax}(\hat{\mathbf{y}}))$ as the ranking loss defined in ListNet. Also note that there are no additional transformer layers needed for TOP. There is however an additional linear unembedding layer $\mathrm{U}_{\text{TOP}} : \mathbb{R}^D \rightarrow \mathbb{R}^V$ in parallel to the NTP head. Both unembedding heads $\mathrm{U}_{\text{NTP}}$ and $\mathrm{U}_{\text{TOP}}$ receive the same hidden state $\mathbf{h}_t^L$, which is the output of the final transformer layer. We refer to these unembedding layers as NTP head and TOP head, respectively. The final loss being optimized is simply a sum of the NTP loss and the TOP loss:

$$\mathcal{L} = \mathcal{L}_{\text{NTP}} + \mathcal{L}_{\text{TOP}} \qquad (11)$$

Through this specific target sequence formulation and ranking loss function, the model is expected to learn an internal representation that can approximately construct the future sequence by returning the most probable order of upcoming tokens. This is expected to be an easier task than trying to exactly predict a future token at some offset.

We find that additional transformer blocks like MTP are not needed for TOP because both the NTP and TOP heads are mainly aligned on the same objective: assigning the highest score to the next token. Although it is possible to train a language model on only the TOP objective, the resulting model will only be able to do greedy generation. An NTP head is still needed for non-greedy, probability sampling-based inference. At inference time, we remove the TOP head and use only the NTP head, making the model equivalent to the original transformer architecture.

### 4.2. Complexity Analysis for Training

Consider an unembedding layer with hidden size $D$ and vocabulary size $V$. When a single token is used as input, the FLOPs for this layer is $2DV$. For MTP, we need an extra transformer block for each future token. Let the number of future tokens to predict be defined as $N$ and the complexity of a standard transformer block is $24D^2$. For $N$ additional future tokens, the total extra computational cost for training MTP models is $(N-1)(24D^2)$. For DS-MTP, we need 2 additional norm layers and 1 linear layer on top of MTP, each requiring $2D^2$ FLOPs. Hence, the total extra computa-

*Table 2.* General language modeling evaluation results of NTP vs MTP vs DS-MTP vs TOP on standard NLP benchmarks. We report the the accuracy (%) and perplexity on Lambada, the accuracy (%) on MMLU (continuation), the normalized accuracy (%) on HellaSwag, ARC (challenge), PIQA, and SciQ, the accuracy (%) on Social IQa, and the exact match score on NaturalQuestions Open and TriviaQA. All benchmarks are evaluated at 0-shot.

| Size | Model | Lambada Acc. ↑ | PPL ↓ | MMLU Acc. ↑ | HS N. Acc. ↑ | ARC N. Acc. ↑ | PIQA N. Acc. ↑ | SciQ N. Acc. ↑ | SIQa Acc. ↑ | Avg. Acc. Acc. ↑ | ΔAcc. vs NTP | NQ Open E.M. ↑ | TriviaQA E.M. ↑ |
|---|---|---|---|---|---|---|---|---|---|---|---|---|---|
| 340M | NTP | 36.35 | 30.34 | 29.81 | 42.53 | 28.84 | 66.65 | 74.90 | **39.82** | 45.56 | | 1.94 | **4.93** |
| | MTP | 35.32 | 35.31 | 29.08 | 42.73 | **29.86** | 66.49 | 77.40 | 39.00 | 45.70 | +0.14 | **2.35** | 2.55 |
| | DS-MTP | 34.66 | 40.99 | 28.47 | 40.29 | 27.56 | 63.76 | 73.70 | 37.92 | 43.77 | -1.79 | 0.36 | 0.87 |
| | TOP | **37.07** | **28.76** | **30.09** | **43.57** | 29.35 | **67.57** | **79.80** | 39.00 | **46.64** | +1.08 | 2.22 | 4.37 |
| 1.8B | NTP | 49.58 | 11.38 | 35.34 | 60.05 | 38.65 | 73.50 | 86.40 | 41.56 | 55.01 | | 4.54 | 11.85 |
| | MTP | 47.93 | 13.69 | 34.76 | 58.29 | 40.61 | 73.07 | 87.20 | 42.12 | 54.85 | -0.16 | 4.46 | 15.98 |
| | DS-MTP | 48.71 | 13.32 | 35.01 | 57.48 | 40.44 | 71.87 | 86.40 | **42.84** | 54.68 | -0.33 | 4.21 | 12.06 |
| | TOP | **50.34** | **11.19** | **36.21** | **60.45** | **42.32** | **74.16** | **87.90** | 42.53 | **56.27** | +1.26 | **5.37** | **18.93** |
| 7B | NTP | 55.89 | 7.97 | 39.47 | 67.44 | 45.65 | **76.99** | 88.60 | **44.37** | 59.77 | | 7.31 | 24.28 |
| | MTP | 53.13 | 8.99 | 38.14 | 65.85 | 45.56 | 75.73 | 89.30 | 44.11 | 58.83 | -0.94 | 7.40 | 23.36 |
| | DS-MTP | 55.62 | 8.52 | 38.16 | 66.03 | 44.37 | 75.79 | 88.70 | 43.76 | 58.92 | -0.85 | 6.57 | 18.54 |
| | TOP | **57.03** | **7.64** | **39.65** | **68.73** | **46.42** | 76.39 | **91.60** | 43.91 | **60.53** | +0.76 | **7.70** | **30.90** |

*Table 3.* Math model results. We evaluate GSM8K at 5-shot exact match accuracy (%) with flexible extract, and MATH at 4-shot math-verified accuracy (%).

| | 1.8B | | 7B | |
|---|---|---|---|---|
| Method | GSM8K ↑ | MATH ↑ | GSM8K ↑ | MATH ↑ |
| NTP | 39.20 | 13.34 | 53.53 | **21.74** |
| MTP | 38.59 | 15.00 | 50.80 | 19.28 |
| DS-MTP | 2.65 | 3.66 | 7.51 | 6.40 |
| TOP | **45.64** | **16.66** | **55.57** | 20.40 |

*Table 4.* Code model results. We evaluate HumanEval at 0-shot and MBPP at 1-shot, and report both benchmarks in pass@16, pass@32, pass@64 accuracy (%).

| | pass | 1.8B | | 7B | |
|---|---|---|---|---|---|
| Method | @ | HumanEval ↑ | MBPP ↑ | HumanEval ↑ | MBPP ↑ |
| NTP | | 30.41 | 41.34 | **36.21** | 44.64 |
| MTP | 16 | 30.70 | **44.29** | 35.21 | 45.66 |
| DS-MTP | | 20.11 | 27.81 | 24.58 | 39.01 |
| TOP | | **32.35** | 40.71 | 34.48 | **48.46** |
| NTP | | 31.98 | 44.16 | **39.10** | 46.95 |
| MTP | 32 | 32.82 | **46.41** | 38.45 | 48.03 |
| DS-MTP | | 22.28 | 30.33 | 27.01 | 42.13 |
| TOP | | **34.96** | 43.16 | 37.74 | **51.48** |
| NTP | | 33.53 | 46.70 | **42.68** | 49.20 |
| MTP | 64 | 34.76 | **48.20** | 42.50 | 50.10 |
| DS-MTP | | 23.78 | 31.90 | 30.18 | 44.90 |
| TOP | | **38.41** | 44.80 | 41.77 | **54.00** |

tional cost for training DS-MTP models is $(N-1)(30D^2)$. For TOP, we only need 1 extra unembedding layer. So, the total extra computational cost for training TOP models is simply $2DV$.

This makes TOP much more scalable compared to MTP. No additional parameters are needed even when adjusting window size. Specifically, a TOP head requires $DV$ extra parameters, while $N$ MTP heads require $(N-1)(16D^2 + 2D)$ assuming a standard transformer block with MLP hidden size $4D$ and 2 RMSNorms. We summarize these comparisons in Table 1. While the unembedding matrix can be large, the cost of a single unembedding layer gets amortized as the model size scales up. In practice, we use a fused Triton kernel that performs both the unembedding and loss calculation block-wise in one pass, making the overhead minimal. This kernel is a modification to fused linear cross-entropy loss kernels from Yang & Zhang (2024), resulting in the same performance as the non-modified version.

## 5. Experiments and Results

### 5.1. General language modeling

We pretrain Llama-style transformer (Pre-Norm, RoPE, SwiGLU MLP) models for the training methods NTP, MTP,

DS-MTP, and TOP in 3 sizes each: 340M, 1.8B, and 7B. These sizes are an approximate naming scheme; each model of each training method will have slightly different parameter counts. We try to match the parameter count at training time, excluding embedding parameters. This means that by setting the MTP or DS-MTP number of future tokens to 4, the shared trunk will be reduced by 3 layers to account for the added MTP heads, as is done in the original MTP paper. We pretrain all models on the sample-100BT subset of FineWeb-Edu (Lozhkov et al., 2024). The 340M models are trained on 52B tokens, while the 1.8B and 7B models are trained on 104B tokens. We use the Flame framework (Zhang & Yang, 2025) and flash-linear-attention repository (Yang & Zhang, 2024) to implement and train our models. The full training configuration and hyperparameters for all model sizes are detailed in Appendix A, Table 9 which originally came from Yang et al. (2024) along with hyper-

parameters for vanilla attention. We set the TOP window size to be as large as possible but still tractable to compute the target sequence with. Here, we set it to be equal to the sequence length, which means Algorithm 1 will receive an input with twice the sequence length.

We evaluate our models on nine standard NLP benchmarks: ARC (Challenge) (Clark et al., 2018), Lambada (Paperno et al., 2016), PIQA (Bisk et al., 2020), SciQ (Welbl et al., 2017), Social IQa (Sap et al., 2019), TriviaQA (Joshi et al., 2017), NaturalQuestions Open (Kwiatkowski et al., 2019), HellaSwag (Zellers et al., 2019), and MMLU (Hendrycks et al., 2021b;a), with full results presented in Table 2. Across all model sizes, TOP shows overall better performance over MTP, DS-MTP, and the baseline NTP models on most tasks.

Our reproduction of MTP shows smaller MTP models achieve competitive results. This finding complements the original MTP paper, which did not report on models smaller than 7B on the standard NLP benchmarks. Consistent with the original study however, the 7B MTP model underperforms in these tasks. While the MTP paper suggests that it scales effectively on coding tasks, our findings indicate that this scalability does not extend to non-coding tasks. In contrast, our TOP model improves in performance as it scales to 7B and surpasses the 7B NTP and MTP baseline. This suggests that in more general tasks, TOP performs and scales better than MTP. We also do not see an improvement in performance from DS-MTP compared to MTP in our reproduction.

## 5.2. Generative tasks

We also evaluate TOP on generative tasks that require forward thinking, such as math and code. Specifically, we use MATH (Minerva few-shot variant) (Hendrycks et al., 2021c; Lewkowycz et al., 2022) and GSM8K (Cobbe et al., 2021) for math benchmarks. For code, we use HumanEval (Chen et al., 2021) and MBPP (Austin et al., 2021). Our pretrained base models are not sufficiently capable for these tasks, therefore we continue pretraining them on more math and code texts. We use 20B tokens from the Python subset of Stack-Edu (Allal et al., 2025) for the code models and 4.2B tokens of OpenMathInstruct-2 (Toshniwal et al., 2024) for the math models. Continued training hyperparameters are detailed in Appendix A, Table 10.

After continued training, the result is 16 additional models pretrained and continually trained using NTP, MTP, DS-MTP, and TOP each, in 1.8B and 7B sizes, with one math and one code model each. We report the results of the math models in Table 3 and the code models in Table 4. Our results show TOP models prevail in generative tasks as well. In math, the TOP 1.8B model outperforms all other models by a considerable margin, while the TOP 7B model is ahead in GSM8K and beats MTP and DS-MTP in MATH. In code

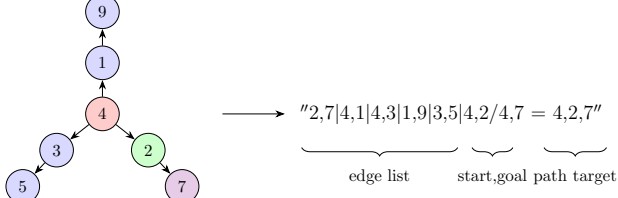

*Figure 3.* Illustration of a star graph training sample with $d = 3$ and $l = 3$ due to Bachmann & Nagarajan (2024).

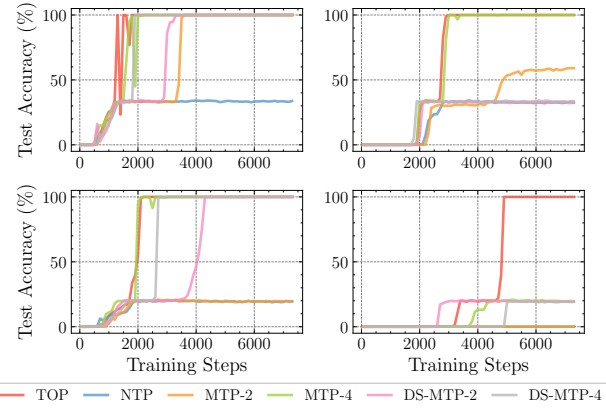

*Figure 4.* Test set accuracy during training of NTP, TOP, MTP, DS-MTP models on the star graph pathfinding task. Star graph setups: $G(3, 3)$ (top left), $G(3, 5)$ (top right), $G(5, 3)$ (bottom left), $G(5, 5)$ (bottom right).

generation, while the results are more mixed, we still see TOP 1.8B and 7B performing best at HumanEval and MBPP respectively. We recognize the unusually low performance of DS-MTP here, and suspect overfitting as the cause, as the base model performs relatively well. The training losses of DS-MTP appear normal as can be seen in the loss plots in Appendix B. We see that the measured NTP training loss of DS-MTP models are usually below MTP models, despite poorer downstream performance.

## 5.3. The star graph task

In addition to general language modeling, we also evaluate TOP on a synthetic task, the star graph pathfinding problem. It was proposed by Bachmann & Nagarajan (2024) to highlight the weakness of training with NTP where the model fails to learn the correct lookahead solution. The task is to find a path from a starting node to a goal node, given a star graph $G(d, l)$ with $d$ paths branching from the starting node and path length $l$. Refer to Figure 3 for an example training sample of this task.

We train standard transformers with 8 layers, embedding size 384, and 6 attention heads for 6 training objectives: NTP, TOP, MTP-2, MTP-4, DS-MTP-2, and DS-MTP-4.

*Table 5.* Parameter count and test accuracy on the star graph task.

| MODEL | PARAMS | $G(3,3)$ | $G(3,5)$ | $G(5,3)$ | $G(5,5)$ |
|---|---|---|---|---|---|
| NTP | 14.2M | 33.8 | 32.5 | 19.5 | 0.1 |
| MTP-2 | 16.0M | **100** | 59.0 | 19.6 | 0.1 |
| MTP-4 | 19.6M | **100** | **100** | **100** | 19.5 |
| DS-MTP-2 | 16.6M | **100** | 32.5 | **100** | 19.2 |
| DS-MTP-4 | 20.7M | **100** | 33.6 | **100** | 19.3 |
| TOP | 14.2M | **100** | **100** | **100** | **100** |

*Table 6.* Average accepted tokens per forward pass for self-speculative decoding. Higher is better.

| MODEL | SIZE | DOMAIN | | | |
|---|---|---|---|---|---|
| | | WIKI | BOOKS | CODE | MATH |
| MTP | 340M | 2.15 | 1.88 | 2.23 | 2.21 |
| | 1.8B | 2.27 | 2.02 | 2.38 | 2.42 |
| | 7B | 2.39 | 2.09 | 2.46 | 2.49 |
| DS-MTP | 340M | 2.81 | 2.72 | 2.60 | 2.69 |
| | 1.8B | 2.96 | 2.82 | 2.75 | 2.81 |
| | 7B | **3.03** | **2.91** | **2.92** | **2.97** |
| TOP | 340M | 1.38 | 1.33 | 1.37 | 1.37 |
| | 1.8B | 1.47 | 1.41 | 1.49 | 1.51 |
| | 7B | 1.52 | 1.42 | 1.53 | 1.55 |

The MTP and DS-MTP models have different number of future tokens i.e. two and four MTP heads. Note that in the case of MTP and DS-MTP in this task, we only subtract one layer from the main trunk and add the MTP heads on top because the models are too small, resulting in MTP models with larger parameter counts compared to NTP and TOP. We train these models on 4 star graph setups: $G(3,3)$, $G(3,5)$, $G(5,3)$, and $G(5,5)$ with $N = 30$ which means each node label is sampled uniformly from 30 labels i.e. tokens. In each setup we generate 300,000 training samples and 10,000 test samples and train for 100 epochs to convergence. We use a batch size of 4096, learning rate of 0.003 with warmup of 1500 and cosine decay down to a learning rate of 0.001 for all models.

We present the results of the star graph task in Table 5 and Figure 4. Similar to the results in (Bachmann & Nagarajan, 2024), NTP performs poorly across the board and fails to learn the appropriate lookahead mechanism. We also observe that the effectiveness of MTP is dependent on its number of heads. For instance, a configuration with four heads is effective, while one with only two is insufficient, particularly for graphs with longer paths. However, the TOP model demonstrates the most robust performance. On the $G(5,5)$ star graph, the TOP model is the only model with perfect test set accuracy, where even the MTP and DS-MTP model with 4 heads fail.

### 5.4. Self-speculative decoding

Speculative decoding (Stern et al., 2018) is a technique for accelerating inference by first using a smaller model to gen-

erate predictions and then employing the original model as a validator. In MTP, all of the heads can be used simultaneously to predict future tokens. The predicted tokens are then validated in a second forward pass using the same model. The technique is referred to as self-speculative decoding since inference and validation are done using the same model. Although TOP is intended to be used only at training time to improve learning, we also explore the possibility of using the TOP head for self-speculative decoding. To do this, we construct a future sequence by ordering the TOP head's predicted tokens by proximity score, appending them to the input, and running another forward pass to check the longest common prefix i.e. acceptance rate with the NTP head's predictions.

We take all TOP, MTP, and DS-MTP models and evaluate their self-speculative decoding potential on texts of different domains. For each domain, we take 5000 random snippets of text and calculate the average number of accepted tokens per validation forward pass. We present the results in Table 6. Evidently, TOP does not perform as well as MTP nor DS-MTP for self-speculative decoding. The 7B TOP model observes a maximum acceptance rate of 1.54. Meanwhile the 7B MTP model goes up to 2.49 acceptance rate, still slightly below the numbers reported in the original paper given 4 MTP heads, while the DS-MTP 7B model achieves up to an impressive 3.03 acceptance rate. We suspect the lack of self-speculative decoding performance is due to TOP not being able to predict repetitive tokens sequentially. This remains an open problem.

### 5.5. Exploring other configurations

#### 5.5.1. WINDOW SIZE

We investigate the effect of varying window size when training using the TOP objective. We pretrain 340M parameter models with the same setup as in Section 5.1, only changing the TOP window size with values 4, 16, 128, 1024, and 4096. We report the downstream benchmark results in Table 7, comparing each setup to the baseline NTP model as well. We observe varying results in each benchmark. We hypothesize that every task might benefit from different amounts of lookahead. All window sizes however outperform the NTP baseline. When considering TOP for practical use, we suggest tuning the window size beforehand on the model and data being used. Unlike MTP though, using a larger window size does not require the trade-off of more parameters.

#### 5.5.2. LOSS WEIGHTING RATIO

We also explore changing the ratio between the NTP loss and the TOP loss when adding them together for the training loss. Specifically, we parameterize the combined loss as $\mathcal{L} = (1 - \alpha)\mathcal{L}_{\text{NTP}} + \alpha\mathcal{L}_{\text{TOP}}$ where $\alpha$ controls the relative

*Table 7.* Effect of window size on 340M TOP models. We report the accuracy and perplexity on Lambada, the normalized accuracy on HellaSwag, ARC (challenge), PIQA, and SciQ, and the exact match score on NaturalQuestions Open and TriviaQA. Best scores are bolded.

| BENCHMARK | WINDOW SIZE | | | | | NTP |
| --- | --- | --- | --- | --- | --- | --- |
| | 4 | 16 | 128 | 1024 | 4096 | |
| LAMBADA ↑ | 37.36 | **38.68** | 37.98 | 36.95 | 37.07 | 36.35 |
| LAMBADA ↓ | 27.42 | **25.30** | 26.69 | 27.80 | 28.76 | 30.34 |
| HELLASWAG ↑ | 43.22 | 43.43 | **43.91** | 43.74 | 43.57 | 42.53 |
| ARC ↑ | 29.78 | **30.55** | 28.50 | 30.12 | 29.35 | 28.84 |
| PIQA ↑ | 66.81 | 68.66 | **69.04** | 67.85 | 67.57 | 66.65 |
| SCIQ ↑ | 77.40 | 75.70 | 78.10 | 76.60 | **79.80** | 74.90 |
| NQ OPEN ↑ | **3.05** | 2.08 | 2.22 | 2.66 | 2.22 | 1.94 |
| TRIVIAQA ↑ | **6.38** | 3.72 | 4.07 | 4.15 | 4.37 | 4.93 |

*Table 8.* Effect of loss weighting ratio $\alpha$ on 340M TOP models, where $\mathcal{L} = (1 - \alpha)\mathcal{L}_{\text{NTP}} + \alpha\mathcal{L}_{\text{TOP}}$. We report the accuracy and perplexity on Lambada, the normalized accuracy on HellaSwag, ARC (challenge), PIQA, and SciQ, and the exact match score on NaturalQuestions Open and TriviaQA. Best scores are bolded.

| BENCHMARK | LOSS WEIGHTING RATIO ($\alpha$) | | | | | NTP |
| --- | --- | --- | --- | --- | --- | --- |
| | 0.10 | 0.25 | 0.50 | 0.75 | 0.90 | |
| LAMBADA ↑ | 36.37 | 36.23 | 37.07 | 37.63 | **38.39** | 36.35 |
| LAMBADA ↓ | 29.27 | 31.00 | 28.76 | **27.12** | 27.17 | 30.34 |
| HELLASWAG ↑ | 43.16 | 43.62 | 43.57 | **44.33** | 44.07 | 42.53 |
| ARC ↑ | 28.75 | 28.92 | 29.35 | 29.86 | **30.72** | 28.84 |
| PIQA ↑ | 67.68 | **68.66** | 67.57 | 67.85 | 67.63 | 66.65 |
| SCIQ ↑ | 76.30 | 77.40 | 79.80 | 79.00 | **80.00** | 74.90 |
| NQ OPEN ↑ | 2.08 | 2.55 | 2.22 | 2.58 | **3.05** | 1.94 |
| TRIVIAQA ↑ | 3.70 | 2.44 | 4.37 | 5.02 | **5.94** | 4.93 |

weight of the TOP objective. We pretrain 340M parameter models with the same configuration as in Section 5.1, varying $\alpha \in \{0.1, 0.25, 0.5, 0.75, 0.9\}$. Surprisingly, the results in Table 8, with comparison to the baseline NTP model, show that higher TOP loss weighting generally improves performance, with $\alpha = 0.9$ achieving the best results on most benchmarks. This suggests that the TOP objective provides a stronger learning signal than initially expected, and that our default equal weighting ($\alpha = 0.5$) may be conservative. In practice, we also suggest tuning the loss weighting ratio beforehand.

# 6. Related Work

## 6.1. Language Model Losses

Many previous works have explored variations of the language modeling loss, most of them for use in training encoder models. Masked language modeling (MLM) randomly masks input tokens and trains the model to recover them from bidirectional context (Devlin et al., 2019). For example, T5 uses a denoising span-corruption objective in which contiguous spans are replaced by sentinel tokens and the model must reconstruct the spans (Raffel et al., 2020); this yields shorter target sequences and faster training. XL-Net's permutation language modeling samples random autoregressive orderings to capture bidirectional dependencies while remaining autoregressive (Yang et al., 2019). Similarly, denoising autoencoder pretraining as in BART corrupts text (e.g., by shuffling sentences or infilling masked spans) and learns to reconstruct the original text (Lewis et al., 2020). UL2 (Tay et al., 2023) further unifies these ideas with a mixture-of-denoisers objective, interleaving various span and prefix corruption schemes to improve robustness across tasks. Retrieval-augmented models like RETRO add a nearest-neighbor retrieval step during pretraining, conditioning generation on retrieved document chunks to reduce perplexity and enable easy knowledge updates (Borgeaud et al., 2022). Other alternatives include replaced-token de-

tection as in ELECTRA (Clark et al., 2020), where the model sees plausible substitutes in place of masked tokens and must identify which tokens were replaced.

## 6.2. Multi-Token Prediction

Next-token prediction has been shown to limit long-range planning due to teacher forcing. The teacher forcing approach may fail to learn an accurate next token predictor, which hinders the model's ability to plan beyond several tokens (Bachmann & Nagarajan, 2024). This issue motivates the exploration of alternative or auxiliary training objectives. Multi-token prediction (MTP) addresses this by jointly predicting multiple future tokens, improving lookahead and planning performance (Gloeckle et al., 2024). MTP has been adopted in recent large models such as DeepSeek V3 (DeepSeek-AI et al., 2024) and Ling-V2 (inclusionAI, 2025), and can also enable faster inference by self-speculative decoding.

The MTP framework has several variants. DeepSeek V3 uses a sequential prediction mechanism with a small lookahead window (N=3) to enhance decoding efficiency (DeepSeek-AI et al., 2024). Other approaches for multi-token awareness include converting NTP models to MTP models using register tokens (Gerontopoulos et al., 2025) and exploring parallel reasoning in a continuous space (Gozeten et al., 2025). Ahn et al. (2025) proposes to predict the joint probability of future tokens by carefully bottlenecking the architecture of the MTP heads. Yin et al. (2024) introduce Semformer and but also evaluate a bag-of-words (BoW) auxiliary objective that predicts the unordered bag of target tokens, although BoW is unable to acquire perfect accuracy. Frydenlund (2025) studies the star graph task through "supervision adulteration" and proposes alternative sequential prediction losses, including a ranking-based multi-token objective that orders future target tokens with a pairwise hinge ranking loss. Finally, Mahajan et al. (2025) proposed Future Summary Prediction (FSP) of 2 variants,

BoW and RevLM. Their FSP-BoW variant is similarly an unordered bag-of-words prediction. TOP differs from these works as it is a general LLM pretraining objective which ranks vocabulary tokens by proximity within a window using a listwise ranking loss, requires only an additional linear unembedding head, and is evaluated at scale on general NLP, math, code, and star graph benchmarks.

## 7. Limitations

Due to limitations in compute, we are unable to pretrain on more data or larger models. The 7B models require 2 weeks of training time each on the 8xH200 node available to us. While our work demonstrates the potential of token order prediction for LLM pretraining, it remains to be seen whether TOP scales well to the standard of larger models and longer training runs of today. We note that the DS-MTP models are a reproduction baseline under our setup and we avoid drawing strong conclusions from its underperformance. Our DS-MTP reproduction may not fully reflect the best achievable performance of the method. However, our main conclusions are primarily supported by the comparisons against NTP and MTP as well as the results in the star graph task.

## 8. Conclusion

In this paper, we propose token order prediction (TOP) as a novel auxiliary training loss for LLM pretraining. Our approach addresses some limitations of multi-token prediction (MTP) by replacing the difficult task of exact future token prediction with the more tractable objective of ranking upcoming tokens by their proximity. TOP requires only a single additional unembedding layer compared to MTP's multiple transformer layers, making it more parameter-efficient and scalable.

Based on the results of our general language modeling experiments across three model sizes (340M, 1.8B, and 7B parameters), TOP overall improves performance over NTP, MTP, and DS-MTP on standard NLP benchmarks. The method shows positive gains as parameter count grows, suggesting its potential value for larger-scale language models. Additionally, TOP also improves performance on coding and math tasks, implying that TOP induces models with better understanding of tasks that require forward thinking. Lastly, we further verify the power of the TOP objective by evaluating on the synthetic star graph pathfinding task. Here, TOP learns the correct lookahead solution on graphs where NTP, MTP, and DS-MTP do not. Although not as effective as MTP or DS-MTP for self-speculative decoding, these preliminary results indicate that TOP offers another promising direction for improving language model training through effective auxiliary objectives.

## Impact Statement

This paper presents work whose goal is to advance the field of language modeling. There are many potential societal consequences of our work, none which we feel must be specifically highlighted here.

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

# A. Training Configuration and Hyperparameters

We present the configurations and hyperparameters used for training all models in Table 9 and hyperparameters used for continued training on math and code in Table 10 below. The architecture configurations for all model sizes are taken from the Flame framework (Zhang & Yang, 2025). These configurations are based on well-established settings from prior work, such as DeltaNet (Yang et al., 2024), which ensures that the reported gains are not due to arbitrary tuning. The specific optimization hyperparameters such as learning rate were taken from the Pythia suite (Biderman et al., 2023) as it contains well-tuned models with similar sizes as ours.

*Table 9.* Training configuration and hyperparameters for {340M, 1.8B, 7B} base models.

| MODEL ARCHITECTURE | |
| --- | --- |
| HIDDEN SIZE | {1024, 2048, 4096} |
| NUM. LAYERS | {24, 32, 30} |
| NUM. HEADS | {16, 32, 32} |
| NUM. KV HEADS | {16, 32, 8} |
| SEQ. LENGTH | 4096 |
| RoPE $\theta$ | 10,000 |
| VOCAB SIZE | 32,000 |
| TIED EMBEDDINGS | FALSE |
| TOP WINDOW SIZE | 4096 |
| MTP/DS-MTP FUTURE TOKENS | 4 |
| **OPTIMIZATION** | |
| OPTIMIZER | ADAMW |
| LEARNING RATE | {3E-4, 2E-4, 1.2E-4} |
| LR SCHEDULE | COSINE (10% MIN) |
| WARMUP STEPS | {1K, 2K, 2K} |
| GLOBAL BATCH SIZE | 128 |
| TRAINING STEPS | {100K, 200K, 200K} |
| GRADIENT CLIP | 1.0 |

*Table 10.* Continued training hyperparameters for {1.8B, 7B} math and code models.

| HYPERPARAM. | MATH | CODE |
| --- | --- | --- |
| OPTIMIZER | ADAMW | ADAMW |
| LEARNING RATE | {3E-5, 2E-5} | {5E-5, 2E-5} |
| LR SCHEDULE | COSINE (10% MIN) | COSINE (10% MIN) |
| WARMUP STEPS | 2K | 400 |
| GLOBAL BATCH SIZE | 128 | 128 |
| TRAINING STEPS | 8K | 40K |
| GRADIENT CLIP | 1.0 | 1.0 |

## B. Pretraining and Continued Training Loss

We present the training losses for each model. The configurations and hyperparameters are detailed in Appendix A.

For the TOP architecture, we report three losses: (1) NTP loss from the NTP head (Equation 1), (2) TOP loss (Equation 10), and (3) total loss, which is the sum of both NTP and TOP losses (Equation 11.

For the MTP and DS-MTP architectures, we report two losses: (1) measured NTP loss, representing the loss from the first head of MTP or DS-MTP heads (Equation 3), and (2) total loss, which is the sum of losses from all heads.

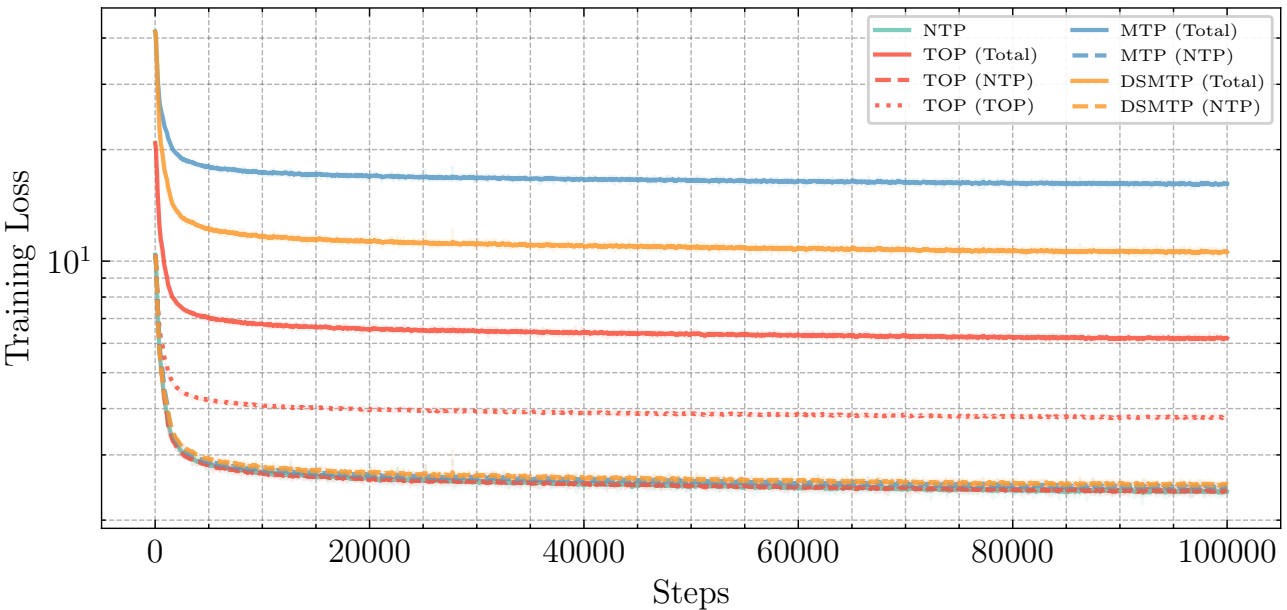

*Figure 5.* Pretraining loss of 340M parameter base models.

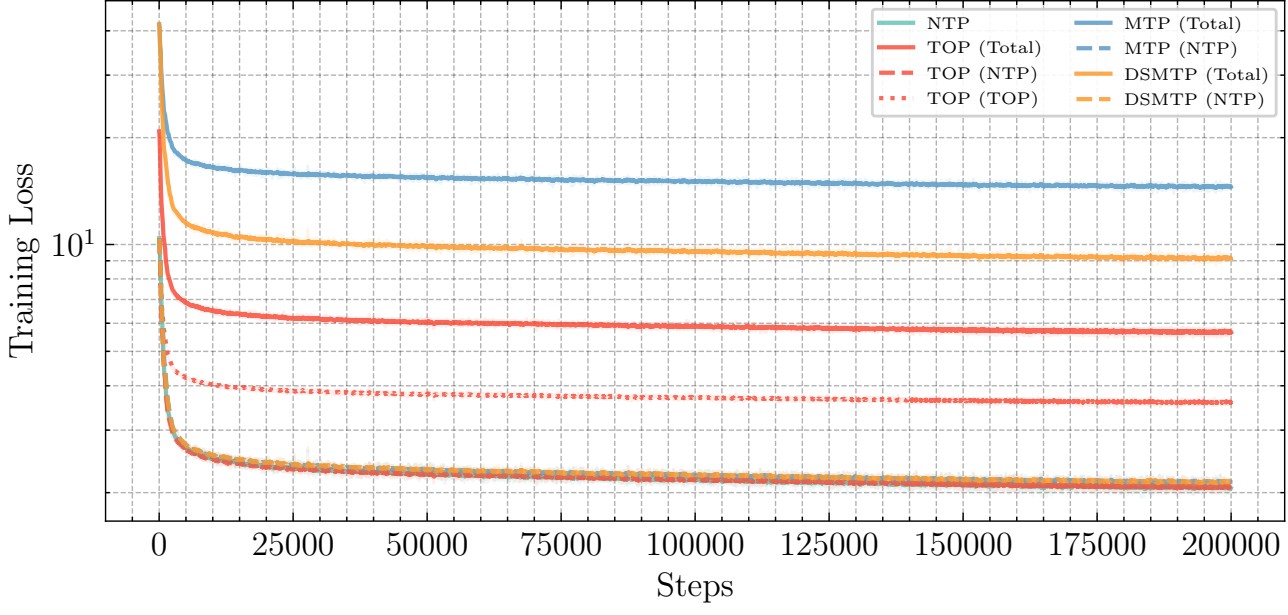

*Figure 6.* Pretraining loss of 1.8B parameter base models.

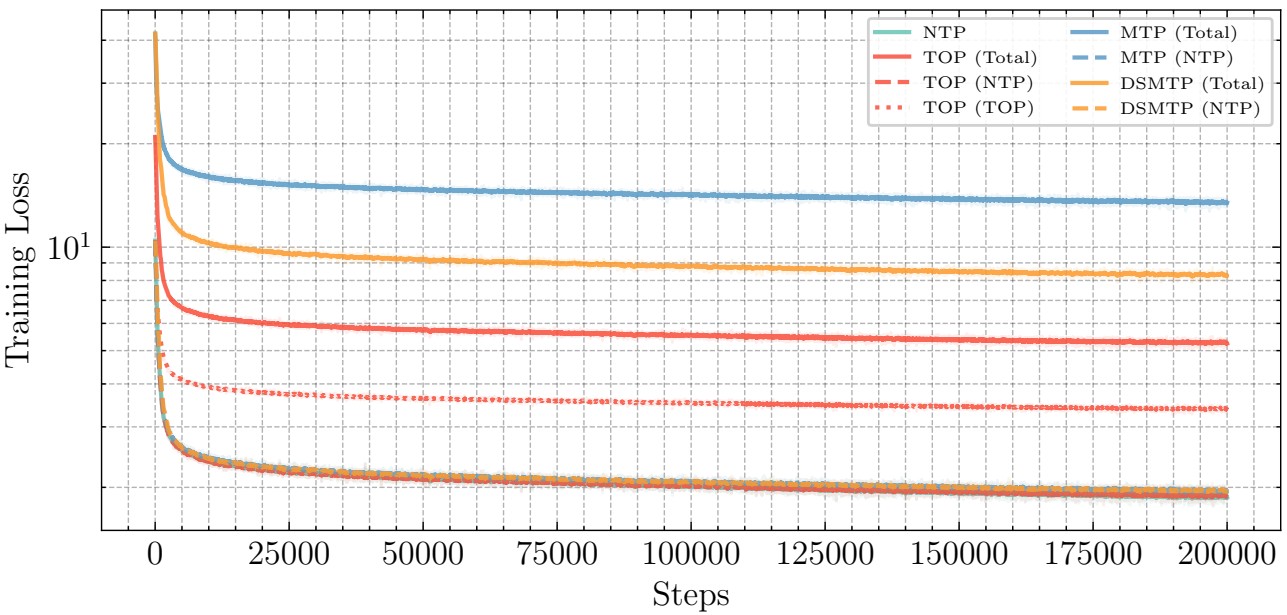

*Figure 7.* Pretraining loss of 7B parameter base models.

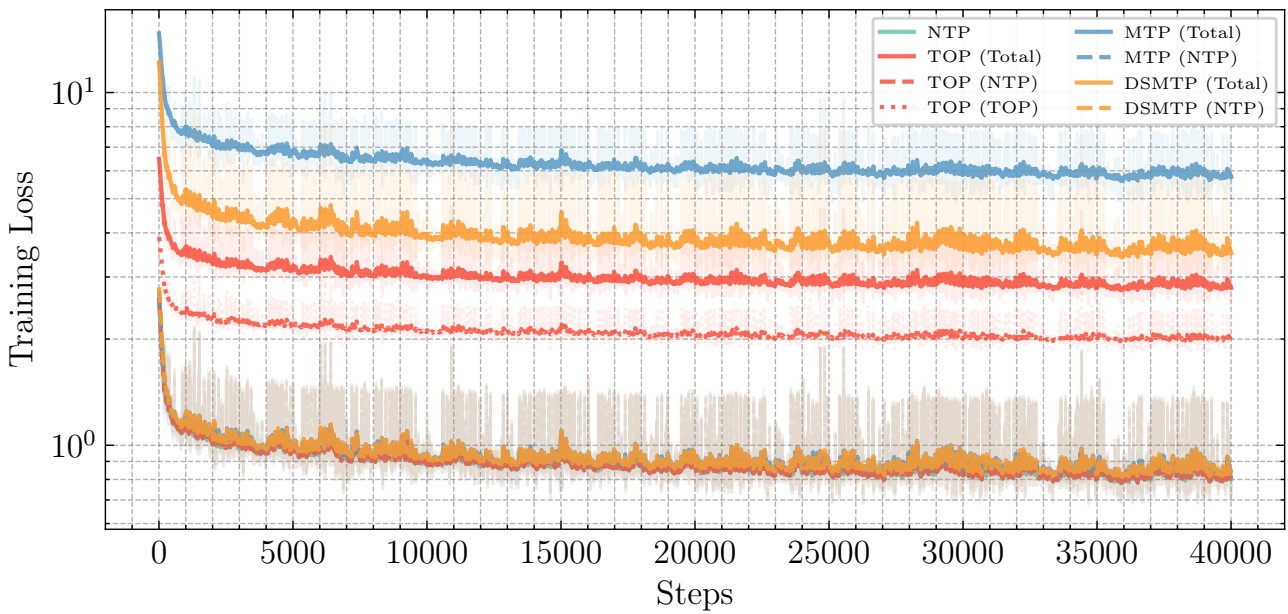

*Figure 8.* Continued training loss of 1.8B parameter models on code data.

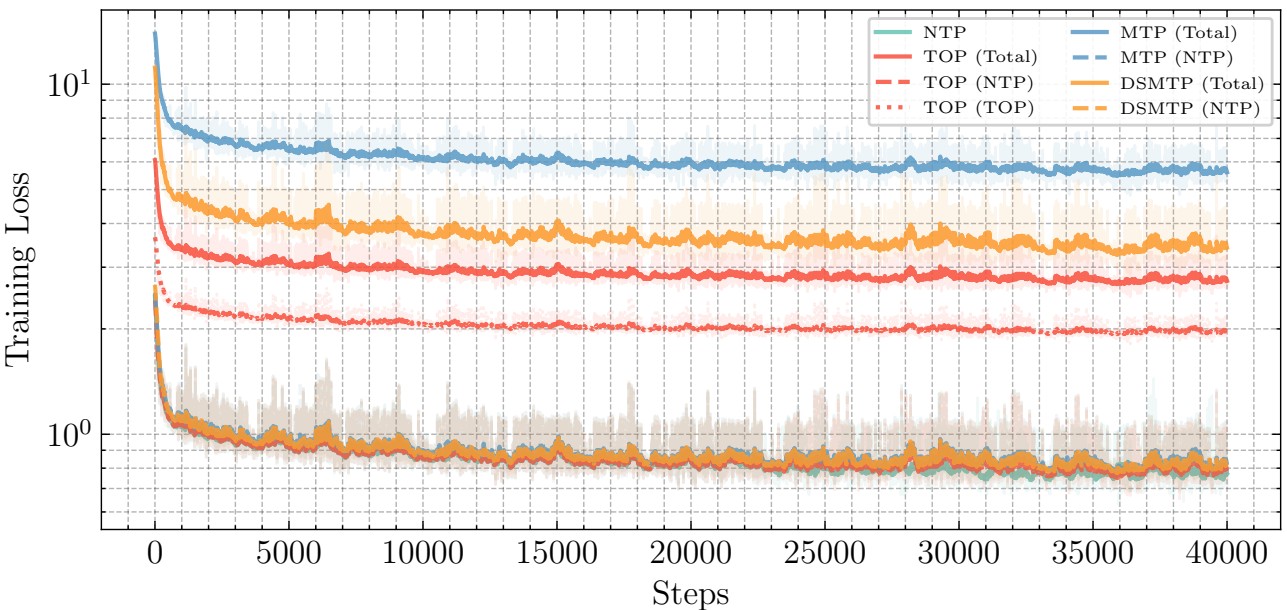

*Figure 9.* Continued training loss of 7B parameter models on code data.

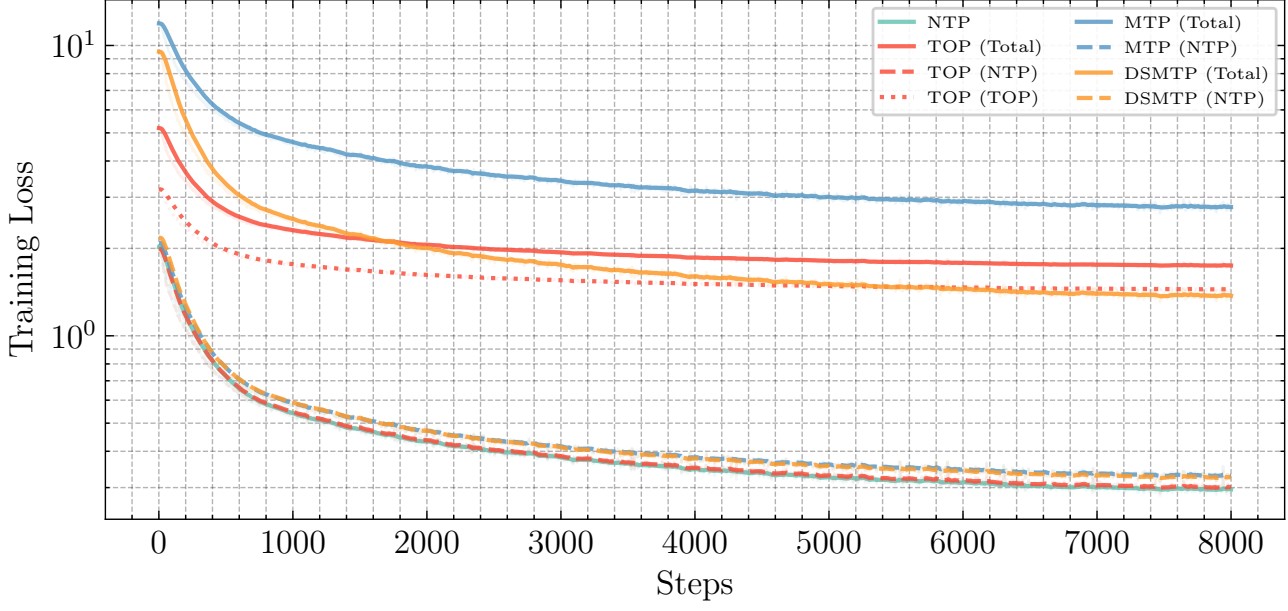

*Figure 10.* Continued training loss of 1.8B parameter models on math data.

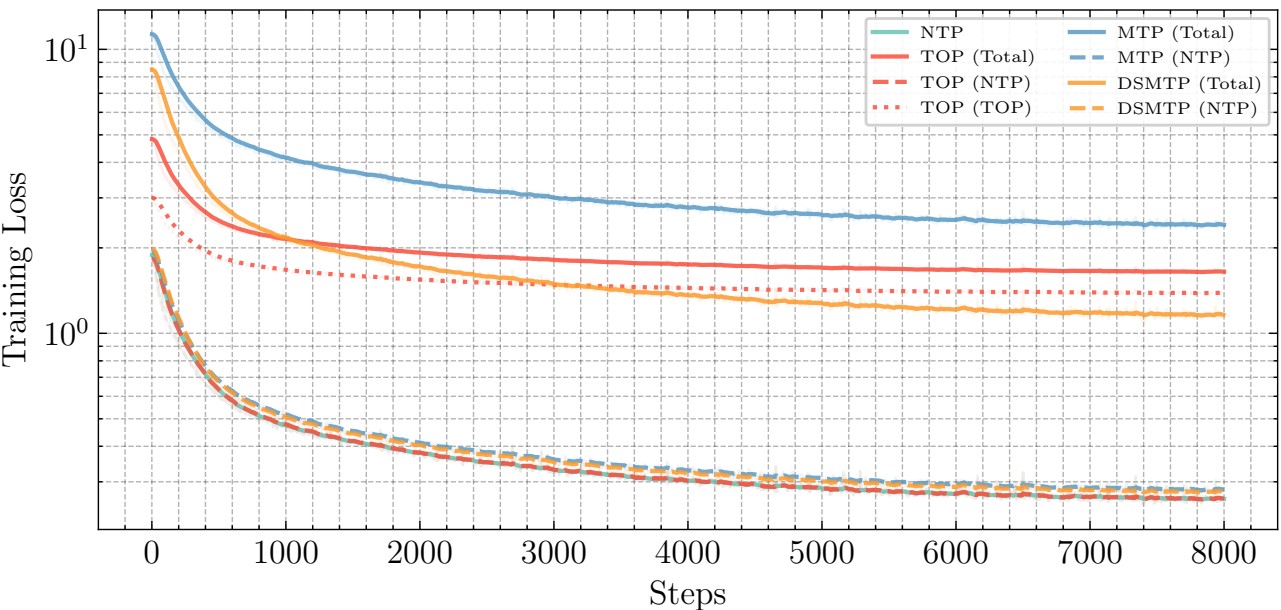

*Figure 11.* Continued training loss of 7B parameter model on math data.

