# OpenReview forum: "Predicting the Order of Upcoming Tokens Improves Language Modeling"
_ICML.cc/2026/Conference — ICML 2026 regular_

### Official Review · Reviewer_NNtG · 2026-03-02

**Soundness:** 3
**Presentation:** 3
**Significance:** 3
**Originality:** 3
**Overall Recommendation:** 4
**Confidence:** 1

**Summary:**

This paper proposes Token Order Prediction (TOP) as a lightweight auxiliary pretraining objective that predicts the relative order of tokens appearing in a future window (rather than exact future tokens as in MTP), implemented via an extra unembedding head, and shows more consistent gains on language modeling and downstream benchmarks across model scales.

**Compliance With Llm Reviewing Policy:**

Affirmed.

**Key Questions For Authors:**

I wonder how well TOP would work in a more practical setting where we start from an already-pretrained base model, since most of the community rarely trains from scratch. It would be very helpful to see an incremental experiment that adds TOP during continued pretraining or post-training on top of an existing open-source model under a comparable compute budget, to better assess deployability and real-world relevance.

**Limitations:**

yes

**Strengths And Weaknesses:**

## Strengths
1. The paper identifies the instability of MTP on general NLP benchmarks and backs the intuition with empirical observations that predicting farther-ahead tokens is substantially harder (higher/slower-decreasing loss), motivating a more learnable auxiliary objective.
2. TOP only adds a parallel unembedding head and an auxiliary loss, avoiding extra transformer heads required by MTP-style methods, which keeps implementation and training cost more controllable.
3. The authors train 340M, 1.8B, and 7B models, compare against NTP/MTP/DS-MTP, and report overall more stable improvements on multiple general benchmarks as well as math and code evaluations.

## Weaknesses
1.   TOP appears sensitive to the window size $W$ and loss weight $\alpha$. While the paper provides ablations and uses $W{=}4096$ in the main setup, it does not offer an automatic selection method or consistent cross-scale guidelines, which may hinder reproducibility and practical adoption.
2. Beyond training benefits, MTP-style objectives are often attractive because they can enable speculative decoding for faster inference. However, the paper’s self-speculative decoding results show substantially lower acceptance for TOP than for MTP/DS-MTP, suggesting that replacing MTP with TOP may trade off the inference-speed gains that MTP can provide.

---

> ### Author Rebuttal · Authors · 2026-03-28
>
> Thank you very much for your thorough review of our paper. We attempt to address your concerns and answer your questions below:
>
> ## W1:
>
> Thank you for raising this concern. We agree that the performance of TOP is affected by the window size 𝑊 and loss weight 𝛼  and that clearer guidance would improve reproducibility and practical adoption. We would like to emphasize two points. First, even without exhaustive hyperparameter tuning, TOP consistently outperforms MTP in our experiments, suggesting that its gains are not limited to a narrowly optimized setting. Second, this sensitivity is not unique to TOP. related methods such as MTP also require tuning of auxiliary loss weights. For example, the K-EXAONE Technical Report (Choi et al., 2026) uses a loss weight of 0.05 for MTP. We will revise the paper to clarify these trade-offs and provide more practical guidance on choosing 𝑊 and α across settings.
>
> ## W2:
>
> Thank you for raising this problem. Indeed, right now the main advantage of TOP vs MTP is that TOP is the better choice for auxiliary loss for better performance since TOP also performs better on general benchmark compared to NTP, unlike MTP that is limited to math and coding gains. We believe that the reason TOP lacks in speculative decoding is that TOP cannot predict repetitive tokens sequentially, therefore suffering in acceptance rate when such tokens repeat within the window. This remains an open problem.
>
> ## Q1:
>
> Thank you for the suggestion. We agree that evaluating TOP after the pretraining stage would be an interesting direction. We believe that TOP may also provide benefits during continued pretraining. At the same time, we note that the primary focus of converting an NTP model to MTP like in Composer 2 (Cursor Research, 2026) is self-speculative decoding for inference. In that setting, TOP performs less favorably than MTP. However, this does not preclude TOP from being useful as an auxiliary training objective for improving overall model performance. Understanding how to improve the acceptance rate of self-speculative decoding with TOP remains an open question.

---

> > ### Author Rebuttal · Reviewer_NNtG · 2026-04-01
> >
> > Thanks for the response, which has largely addressed my concerns.

---

> > > ### Author Response · Authors · 2026-04-07
> > >
> > > Thank you very much for your thoughtful follow-up. We are glad that our response addressed your concerns, and we sincerely appreciate your positive assessment.

---

### Official Review · Reviewer_FWSc · 2026-03-09

**Soundness:** 2
**Presentation:** 3
**Significance:** 3
**Originality:** 3
**Overall Recommendation:** 4
**Confidence:** 3

**Summary:**

This paper proposes Token Order Prediction, TOP, a new auxiliary training objective for large language models. The authors argue that MTP introduces a difficult objective, and therefore hurting performance. Instead, TOP predicts the token orders, which is a relatively easier task, improving LLMs performance. TOP is evaluated on NLP, math, and coding tasks, and the comparison results against NTP, MTP, DS-MTP validate its superior performance.

**Compliance With Llm Reviewing Policy:**

Affirmed.

**Key Questions For Authors:**

1. Please refer to the weaknesses. A more in-depth analysis of why MTP fails, and the mechanism of TOP (e.g., does TOP capture true long-range semantic dependencies?) will strength the paper. Also, though authors mention limitations in compute, if TOP proves effective for larger models, it will be a big plus.
2. For star graph task, MTP is configured of using 2 or 4 heads, what is the window size of TOP? will a large window size incur an unfair compasiron? Also, in Table 6, the results are mixed for different window size, so inpractice, how to set a proper window size?
3. This paper has many issues for references, i.e., incomplete references. For example, the first reference lacks venue information, and some arxiv papers miss IDs.

**Limitations:**

The limitation of unable to pretrin larger models is discussed.

**Strengths And Weaknesses:**

Strengths:
1. The main idea of predicting the order of future tokens rather than their exact identities is a novel, and well-motivated departure from MTP. The authors claim that exact future token prediction is a difficult task, which leads to inconsistent performance, and propose a more tractable objective.
2. The method is evaluated on various model scales (340M, 1.8B, 7B), different tasks (NLP, math, coding, and synthetic star graph task).
3. The authos also provide a complexity/efficiency analysis for TOP.

Weaknesses:
1. This paper is built on the core assumption: MTP is too difficult as a learning objective. But the paper lacks a deeper analysis of why TOP is more effective than MTP. Is it purely because the task is easier? Or does the ranking loss provide a denser, more informative gradient? Also, as noted in the paper that a certain capability threshold is required for MTP’s multi-token modeling, but for a more capable model, will a harder learning objecttive enhance a model's capability ceiling？In other words, if the base models exceed some threshold, will MTP be better than TOP?
2. DS-MTP significantly underperform other approaches for math and code tasks, and the authors explain that it may be due to overfitting.  This may raise the concern of reproduction reliability. A deeper investigation is needed on why DS-MTP fails on these tasks.

---

> ### Author Rebuttal · Authors · 2026-03-28
>
> Thank you very much for your thorough review of our paper. We attempt to address your concerns and answer your questions below:
>
> ## W1:
>
> Thank you for raising this point. We agree that the current paper does not fully explain why TOP is more effective than MTP. We will revise the discussion to make this limitation clear. Our current view is that TOP likely benefits from two factors at once: a simpler target and a denser supervision signal.
>
> First, TOP is an easier task than exact multi-token prediction. It asks the model to predict a coarser property of the future sequence, namely the relative order or proximity of upcoming tokens, instead of the exact token at each offset. Because of this, multiple future windows can map to the same TOP target. TOP can also give partial credit to a near miss. In contrast, MTP uses one-hot cross-entropy, where every wrong prediction is treated the same.
>
> Second, the ListNet objective may provide a denser gradient than MTP. MTP supervises one exact token for each offset. TOP instead produces a soft target distribution over several relevant tokens inside the look-ahead window. This may make optimization less brittle, especially for smaller or less capable models where exact future-token prediction is hard. At the same time, we do not claim that MTP is always worse than TOP. It is possible that, beyond the scales studied here, exact future prediction may become more useful. Moreover, right now TOP is only better as an auxiliary loss that helps the performance of the model. To use the component as self-speculative decoding, MTP still a better choice. We see this as an open question.
>
> We thank the reviewer for pointing out this important direction for deeper analysis.
>
> ## W2:
>
> We thank the reviewer for raising this concern regarding our DS-MTP reproduction scores. We offer two main pieces of evidence in support of our implementation's correctness: a) the pretrained base model's performance without fine-tuning is consistent with the other models, as shown in Table 2, and b) the loss curves in Appendix B for both the NTP loss and the MTP/DS-MTP auxiliary losses follow expected behavior. Since all models share the same architecture at inference time where the MTP and TOP heads being discarded, we do not believe the observed performance differences in math and code stem from an implementation error.
>
> ## Q1:
>
> Adding to the stated points above, our main answer to “does TOP capture true long-range semantic dependencies?” is the experimental result of the star graph task, which provably requires long-range/lookahead depend. TOP is also proven to be effective on larger models based on our result on the 7B parameter model, which is an important threshold to medium size models these days.
>
> ## Q2:
>
> Correct, in this setup a large window size adds a considerably large amount of parameters for MTP models. Additionally, the star graph task in theory only requires lookahead of the second token after the root, which even with 2 heads is sufficient. However in practice, more heads are needed for convergence. Similar to MTP models of today with their number of MTP heads, the window size of TOP is a hyperparameter that needs to be determined based on heuristics such as data characteristics and hardware optimization, and may be required to be tuned beforehand.
>
> ## Q3:
>
> Thank you, we will revise this in the next version.

---

> > ### Author Rebuttal · Reviewer_FWSc · 2026-04-01
> >
> > The idea that TOP serves as an auxiliary loss is interesting, and I keep positive score.

---

> > > ### Author Response · Authors · 2026-04-07
> > >
> > > Thank you very much for your thoughtful follow-up and for keeping a positive score. We are glad that our rebuttal addressed your concerns, and we appreciate your encouraging assessment of TOP as an auxiliary loss.

---

### Official Review · Reviewer_Eruu · 2026-03-09

**Soundness:** 3
**Presentation:** 3
**Significance:** 3
**Originality:** 4
**Overall Recommendation:** 4
**Confidence:** 4

**Summary:**

The paper presents Token Order Prediction (TOP) - a novel auxiliary training loss for unsupervised training of generative language models. In TOP, a separate head is trained to predict the future proximity of each vocabulary token relative to the current timestep, up to some lookahead window size. To calculate and optimize this auxiliary objective efficiently, authors utilize ListNet - an established ranking method in document retrieval.

TOP is evaluated together with three baselines across different model sizes on various NLP, coding, and math benchmarks, together with a synthetic benchmark. The new method offers mixed performance, with modest improvements on most tasks and slight degradation on others. Improvements seem to be most pronounced at the 1.8B model size.

Authors also present ablation experiments comparing different window sizes and auxiliary loss weights. Additionally, authors evaluate TOP's ability to perform self-speculative decoding, showing that TOP does not perform well in this task.

**Compliance With Llm Reviewing Policy:**

Affirmed.

**Final Justification:**

My final recommendation mirrors my Strengths And Weaknesses analysis. The rebuttal addressed minor points and I've raised my score on the Significance front.

**Key Questions For Authors:**

1. Do you plan to open-source the training pipeline source code? If yes, this would change my evaluation of the paper on the Significance/Reproducibility front.

**Limitations:**

Some limitations are addressed.

However, I think authors should also address the mixed quantitative results, particularly on coding benchmarks. A potential reason for this might be the mentioned lack of extensive compute resources, prohibiting more thorough tuning of hyperparameters. For example, the ablation in Section 5.5.2 suggests that a possible avenue for improvement could be increasing the weight of the auxiliary loss.

**Strengths And Weaknesses:**

*Note on notation*: By [104l] and [048r], I mean line 104 left and line 48 right, respectively.

**Soundness:**

Performance of TOP together with three baselines is thoroughly evaluated across different model sizes on 9 NLP benchmarks, 2 coding benchmarks, 2 math benchmarks, and a synthetic task. The largest evaluated model has 7B parameters, making the results applicable in real-world scenarios. The reviewer commends authors for including also experiments where TOP did not perform well, such as the self-speculative decoding evaluation.

Areas of improvements:
- The soudness of the evaluation protocol on the GSM8K and MATH benchmarks is seriously contested by the collapsed score of DS-MTP. This should be thoroughly studied and explained if the results should remain in Table 3.
- The soundness of the evaluation protocol on coding benchmarks is contested by the fact that MTP underperforms relative to NTP on HumanEval at the 7B scale, contradicting the results in Gloeckle et al. (2024) where MTP beats NTP.
- The model size in Section 5 is reported excluding embedding parameters [257l]. I assume this also means excluding unembedding parameters. However, TOP adds a new unembedding layer $U_\text{TOP}$, so this methodology decision could potentially be unfairly beneficial for TOP. This should be addressed and justified in the text.
- In the star graph task analysis, Section 5.3 should mention the window size used in TOP.
-  The complexity analysis in Section 4.2 should include the effect of the window size $W$. The training runtime overhead as a function of $W$ should also be measured empirically. While [158r] mentions that Algorithm 1 incurs practically no overhead, [270r] mentions: "We want the TOP window size to be as large as possible but still tractable to compute the target sequence with."
- [178r] "We find that additional transformer blocks like MTP are not needed for TOP because both the NTP and TOP heads are mainly aligned on the same objective: assigning the highest score to the next token." - This appeal to intuition is should be justified in more detail, since the TOP head's objective includes assigning correct ranks to tokens other than the immediately following one. The claim would also benefit from quantitative ablation.

**Presentation:**

The method, training configuration, and evaluation protocols are relatively clearly presented.

Although Algorithm 1 and the TOP loss are simple, they're unnecessarily difficult to understand from the exposition in Section 4.1. Specifically:
- The second sentence of Section 4.1 [132r] should be reformulated, potentially split into more sentences.
- Regarding [150r] and line 4 of Algorithm 1: What does is mean for a token to not be valid? When does that occur?
- It should be noted how the end-of-training-sequence edge case is handled. Are the last $W$ tokens discarded?
- [193l] "This listwise ranking loss is formulated as the distance between the top-one probability of two lists of scores, where the distance metric is crossentropy." - This sentence cannot be understood without reading the ListNet paper. Due to the importance of the loss definition, the underlying ListNet loss should be briefly described in Section 2. If this is not possible, I would recommend adding the phrase "as defined in ListNet" to the mentioned sentence, so that the reader knows the sentence is not meant to be understood in its own.

Other areas of improvement:
- In Figure 2 Right, it makes no sense to show the Averaged MTP Loss, since it is dominated by far-looking heads and greatly depends on the chosen number of MTP heads. I would recommend removing the right side of Figure 2, or keeping only the the TOP Loss.
- In Appendix B, I recommend adding an additional table describing the configuration used in the star graph task.
- To enable reproduction and building on top of the presented method, the training pipeline should be made open source.

Minor points:
- On HumanEval and MBPP, it would be nice to present results at pass@10 so that they can be compared to the Gloeckle et al. (2024).
- [046r] Transformer should be cited.
- [071r] superfluous space before before the colon
- [127l] "We define E : Z → R D as the embedding layer also shared with the main transformer trunk." - This sentence is unclear, the part starting with "also" should be reformulated.
- The references should be clickable.


**Significance:**

The paper addresses an important issue of enhancing the widely used next-token prediction loss. Any improvements in this area are of great practical significance.

However, in order to establish TOP as a viable direction of future research, the authors should either tune their method to yield more convincing quantitative results or make the training code open-source to enable others to build on top of it.


**Originality:**

The paper cleverly augments next-token prediction with established methods from the field of document retrieval.

I would recommend citing the paper "Beyond Multi-Token Prediction: Pretraining LLMs with Future Summaries" in Section 6.2, as it pursues a similar direction, predicting the future occurrence of tokens (but not their order).

---

> ### Author Rebuttal · Authors · 2026-03-30
>
> Thank you very much for your thorough review of our paper. We attempt to address your concerns and answer your questions below, within the limited character limit (S for soundness, P for presentation):
>
> ## S1
>
> We offer two main pieces of evidence in support of our implementation's correctness: a) the pretrained base model's performance without fine-tuning is consistent with the other models, as shown in Table 2, and b) the loss curves in Appendix B for both the NTP loss and the MTP/DS-MTP auxiliary losses follow expected behavior. Since all models share the same architecture at inference time where the MTP and TOP heads being discarded, we do not believe the observed performance differences in math and code stem from an implementation error.
>
> ## S2
>
> We believe we have replicated MTP the best we can at the scale that we are operating at. The main difference being the amount of fine-tuning/continued training data. We were able to use 20B tokens of code, while the MTP paper used 91B tokens. Of course, we will open source our implementation so the community can verify the implementation as well.
>
> ## S3
>
> The 7B TOP model contains 6,936,580,096 parameters while the 7B NTP model contains 6,805,508,096 parameters. We will include this explicitly in the next version of the paper, along with the other model sizes.
>
> ## S4
>
> Indeed, experiment settings are important even on the ablation such as the star graph tasks. For the star graph, we are using the same window size heuristic as our main methodology, which is equal to the sequence length. Since each star graph configuration may have different sequence lengths, the window sizes adjust accordingly. We will clarify this in the paper.
>
> ## S5
>
> You are correct that the original complexity discussion in Section 4.2 was incomplete. Table 1 analyzes only the additional model-side training cost introduced by each auxiliary objective. For TOP, this model-side cost is independent of W, since TOP adds only a single extra unembedding head and ranking loss. However, when TOP targets are generated online during training, there is an additional preprocessing cost that depends on the effective input length T + W. In our implementation, this preprocessing is performed by a custom Triton kernel rather than Python-side preprocessing, which keeps the overhead low in practice, but it is still not strictly independent of W. We will clarify this distinction in the paper.
>
> ## S6
>
> Thank you for the suggestion. We will include an ablation for this architectural decision. Meanwhile, here results from a small setup similar to the one used in Section 3:
> | Model                      | Train Loss | Val Loss |
> |----------------------------|------------|----------|
> | TOP                        | 1.04134    | 1.17413  |
> | TOP w/ Transformer Block   | 1.05678    | 1.18941  |
>
> ## P2
>
> You are right that “valid” was underspecified. In our implementation, a token is valid iff it is not the padding token and its id lies in [0, V-1]. Invalid cases therefore correspond to padded positions or malformed/out-of-range token ids. In our standard pretraining setting, tokens are ordinarily all valid. The conditioning is included for robustness and to support padded inputs. We will clarify this explicitly in Algorithm 1 and the surrounding text.
>
> ## P3
>
> We do not discard this edge case during pretraining. In our setup, the corpus is streamed as a continuous token sequence rather than segmented into isolated sentence-level examples. Each TOP training example is therefore constructed from a span of length T + W, so the final W tokens needed for look-ahead target construction are available from the ongoing token stream.
>
> ## Writing
>
> We greatly appreciate the time and attention devoted to reading the manuscript so thoroughly and to providing specific comments on the paper content, including P1, P4, P5, P6, P8, P9, P10, P11, and P12, as well as the suggestion to cite “Beyond Multi-Token Prediction: Pretraining LLMs with Future Summaries” in Section 6.2. We are grateful for these insightful suggestions, and we will address them carefully in the next revision of the paper.
>
> ## On Open Source
>
> We believe it's important to fully release the training pipeline. We plan to open source the entire end-to-end training stack used in this work. This includes preprocessing, training code, config files, WandB logs, checkpointing, optimizer and scheduler states, and the exact pinned dependency versions we used for the reported experiments. We also use open source pretraining data only. The experiments on the star graph task are also going to be open sourced on a separate repository. We want to make it easy for others to reproduce our results and build on top of them. Additionally, because we are forking from an already established framework (Flame and flash-linear-attention) any future updates can also be implemented into our training pipeline.

---

> > ### Author Rebuttal · Reviewer_Eruu · 2026-04-02
> >
> > Thank you very much for your thorough rebuttal. Your commitment to making certain edits and to open-sourcing your approach resolved most of my concerns.
> >
> >  I'm still concerned about S1 and S2. Even when your implementation of DS-MTP and MTP is correct, the fact that they significantly underperform their original presented scores indicates that their training hyperparameters should be tuned. It's important not to tune hyperparameters for the new presented method, and then use them unchanged for the baselines. Of course, it's possible that the original baseline papers reported incorrect results - however, in such a case, the claim should be made and justified.
> >
> > I keep my positive score.

---

> > > ### Author Response · Authors · 2026-04-07
> > >
> > > Thank you very much for the thoughtful follow-up and for keeping your positive score.
> > >
> > > We agree with this concern. MTP and DS-MTP were reproduced under our shared training setup, but not extensively re-tuned in a method-specific way. We therefore will not treat their weaker performance in our setup as definitive evidence about the methods themselves.
> > >
> > > In the revision, we will make this explicit, avoid strong conclusions from their underperformance, and focus our main conclusions primarily on the gains over NTP, MTP, and the star-graph results.

---

### Official Review · Reviewer_QZTG · 2026-03-13

**Soundness:** 2
**Presentation:** 2
**Significance:** 2
**Originality:** 2
**Overall Recommendation:** 4
**Confidence:** 4

**Summary:**

The authors propose an auxiliary loss for pre-training LLMs: Token Order Prediction (TOP). Rather than predicting exactly the future tokens as done with Multi-Token Prediction (MTP), TOP trains the model to rank upcoming tokens by proximity, akin to ListNet rank loss. They pretrain models of different sizes (340M, 1.8B, 7B) on FineWeb-Edu and evaluate extensively on various NLP benchmarks, math/code tasks and the graph-star pathfinding task. TOP generally outperforms NTP, MTP and DS-MTP baselines across most settings.

**Compliance With Llm Reviewing Policy:**

Affirmed.

**Final Justification:**

The authors' rebuttal resolved my most significant concerns: the architecture is confirmed to be standard softmax attention, the invalid loss comparison in Figure 2 will be corrected, and the provided aggregate metrics confirm consistent improvements over NTP at all scales. The method is clean, parameter-efficient, and the star graph result is compelling. The remaining limitations (modest benchmark gains, unreliable DS-MTP baseline, and lack of theoretical insight) are real but acceptable for an empirical contribution. I raise my score from 2 to 4.

**Key Questions For Authors:**

1. What happens when you use $\alpha=0.9$ at 1.8B and 7B scales? Is the trend from Table 8 consistent at larger scales?
2. Can you verify that the DS-MTP implementation is correct? The results seem too poor and I suspect there could be a bug.
3. Could you try a simpler auxiliary loss (bag of words) to isolate the contribution of the ranking structure?
4. Why does TOP perform so well on the star graph but poorly on self-speculative decoding? Could you provide any insight into what kind of future-token information TOP actually encodes?
5. What attention mechanism do the models use in the main experiments? If using linear attention variants, have you validated that TOP’s benefits transfer to standard softmax attention transformers

**Limitations:**

yes

**Strengths And Weaknesses:**

# Strengths

 1. **Clean and well motivated idea**: The core insight is intuitive and well supported empirically.
 2. **Parameter efficiency**: TOP requires only a single additional unembedding layer (DV parameters) versus MTP's $(N−1)(16D² + 2D)$ parameters from multiple transformer blocks. The complexity analysis in Table 1 makes this advantage concrete. Window size can be adjusted without adding any parameters, unlike MTP where adding heads increases cost linearly.
 3. **Comprehensive experimental scope**: The paper trains 12 base models (3 sizes x 4 methods) plus 16 post-training experiments (4 methods x 2 sizes x 2 domains).
 4. **Star-graph results**: TOP achieves 100% accuracy on all four star graph configurations, including G(5, 5) where every other method fails. This is strong evidence that TOP improves the model’s internal look-ahead representations.

# Weaknesses

1. **Improvements on standard benchmarks are modest and inconsistent**: While the paper claims TOP “overall outperforms” baselines, the per-benchmark results in Table 2 are mixed. Many improvements are small and within noise (e.g. 7B MMLU: 39.65 vs 39.47). The paper would benefit greatly from an aggregate metric (e.g. average normalized score across benchmarks). Multiple seeds for training are likely infeasible at 7B scale given the author’s compute constraints.
2. **DS-MTP reproduction quality is questionable**: the DeepSeek MTP anomalously under-performs. The authors justify this as “overfitting” (line 309) and don’t investigate further. But this raises serious concerns as to whether the DS-MTP implementation is correct.
3. **Window size and loss weighting are explored at 340M only, and the default might be suboptimal**: Section 5.5 explores window size and $\alpha$, but only at 340M scale. This exploration reveals that $\alpha=0.9$ outperforms $\alpha=0.5$ on most benchmarks (Table 8), but the main experiments use equal weighting. This is a significant gap, since the authors cannot claim to have found the right operating point for TOP.
4. **Architecture is unclear**: The paper repeatedly refers to “transformer models” and “standard transformers”, but the implementation details strongly suggest linear attention architectures. They use the Flame framework and the flash-linear-attention package. If the main experiments indeed use linear-attention rather than standard softmax attention, this is a significant omission. The paper should explicitly state the attention mechanism used and ideally include at least one comparison on standard softmax attention.
5. **Figure 2 (right panel) makes an invalid cross-loss-function comparison**: a central piece of the paper’s motivation is that the TOP objective is easier to learn that MTP. To support this claim, the authors include Figure 2, showing in the right panel a comparison between the MTP and the TOP losses during training. They say (lines 125-126) that “Compared to a similarly sized model with the Top objective at window size 16, we see that the TOP loss is lower”. However, these are fundamentally different loss function: the MTP loss is an average of cross-entropy losses against one-hot targets over the vocabulary, while the TOP loss is a ListNet ranking loss betweensoftmaxxed proximity scores and softmaxxed model outputs. Comparing their numerical values is meaningless.
6. **Theoretical understanding is lacking**: why does the ranking upcoming tokens help learn better representations? The paper offers no formal analysis of what information the TOP loss gradient provides to the shared trunk. This gap is also present in the star-graph task, where ToP achieves 100% accuracy on all configurations but the authors never explain why it succeeds where the other baselines fail.
7. **Misisng ablations and comparisons**: The paper does not compare against other auxiliary objectives beyond MTP variants. A simpler baseline such as predicting the *set* of upcoming tokens without ordering (bag of words) would help isolate whether the ranking structure specifically matters or whether any future-aware auxiliary signal suffices.

# Minor issues

- In section 4.1, the readers are directed to Figure 11, but it seems this is the wrong figure and it seems authors were meaning to direct readers to figure 1.

---

> ### Author Rebuttal · Authors · 2026-03-28
>
> Thank you very much for your thorough review of our paper. We attempt to address your concerns and answer your questions below (within the character limit of this rebuttal):
>
> ## W1
>
> We will update the table in the next version of the paper with aggregate metrics. Meanwhile, we present the aggregate metric, specifically the average over normalized accuracy, across benchmarks here. Note that MTP underperforms NTP at 7B, which is in line with the results of the original MTP paper (Gloeckle et al., 2024) where it underperforms on NLP benchmarks outside of math and coding:
>
> | Size | Model | Avg. Acc. ↑ | ΔAcc. vs NTP |
> |------|-------|-------------|--------------|
> | 340M | NTP | 45.56 | -- |
> | 340M | MTP | 45.70 | +0.14 |
> | 340M | DS-MTP | 43.77 | -1.79 |
> | 340M | **TOP** | **46.64** | **+1.08** |
> | 1.8B | NTP | 55.01 | -- |
> | 1.8B | MTP | 54.85 | -0.16 |
> | 1.8B | DS-MTP | 54.68 | -0.33 |
> | 1.8B | **TOP** | **56.27** | **+1.26** |
> | 7B | NTP | 59.77 | -- |
> | 7B | MTP | 58.83 | -0.94 |
> | 7B | DS-MTP | 58.92 | -0.85 |
> | 7B | **TOP** | **60.53** | **+0.76** |
>
> ## W2
>
> Our main proof of validity for our implementation of DS-MTP is a) that the performance of the pretrained base model without fine tuning is in line with the other models as presented in Table 2, and b) that the loss plots seen in Appendix B for both the NTP loss and the MTP/DS-MTP losses are normal. At inference time, all models use the same architecture as the MTP/TOP heads are discarded. Therefore we believe the unusual performance in math and code are not due to an implementation issue.
>
> ## W3
>
> We intentionally utilized equal weighting (α=0.5) for our main large-scale experiments to establish a standardized comparison with the baselines. Our primary goal was to demonstrate the fundamental efficacy of TOP without relying on extensive, sometimes prohibitively expensive hyperparameter optimization. The ablations at the 340M scale were subsequently conducted to explore the method's parameter sensitivity and upper limits. Even so, as our main results show promising performance at the non-optimal setup of TOP, these exploration results show that there could be even more upside in using TOP.
>
> ## W4
>
>
> We would like to clarify a misunderstanding: we are using standard transformers, specifically with a Llama-style (pre-norm, RoPE, SwiGLU MLP) architecture. Although we use the Flame framework and the flash-linear-attention package, we are using standard softmax attention for all of the experiments. The framework simply provides an optimized implementation and provides good performance for multi-GPU training. We will add this clarification in the next version of the paper.
>
> ## W5
>
>
> The main purpose of this section is to visualize the worsening of the MTP loss as the future offset increases, therefore showing its difficulty. While the MTP loss(es) and TOP loss aren’t directly comparable, we observe that empirically the NTP loss (i.e. the MTP model’s first MTP head CE loss vs the TOP model’s NTP head CE loss) is lower for the TOP model than the MTP model by 0.085 in this section’s experimental setup. In the next version of this paper, we will replace the MTP vs TOP loss comparison with a direct TOP NTP vs MTP NTP loss plot instead. We will rephrase this section to be more accurate.
>
> ## W6
>
>
> TOP was built on the main driving idea behind MTP, which is to provide models with lookahead signal during training. As is studied in “Pitfalls of NTP” (Bachman & Nagarajan, 2024), the paper that proposed the star graph task, this can be crucial in making accurate next token predictors learnable. With TOP, the idea and the theoretical motivation remains the same. The difference lies in how the architecture of TOP is more compact than MTP. As for the star graph task, the NTP baseline provably cannot solve it (Bachman & Nagarajan, 2024) while the MTP models fail at longer graph lengths due to larger offsets being more intractable, as per Section 3. We will add this explanation in the next version of the paper.
>
> ## W7
>
> While the idea of a simpler bag-of-words baseline would be compelling to test, it would not exactly isolate the ranking structure of TOP as it would require a different loss function (perhaps MSE) instead of the ListNet loss as in TOP, therefore it would not be an accurate ablation.
>
> ## Q1
>
> We agree that Table 8 may suggest a broader conclusion which is supported by the experiments. We only evaluated the TOP/NTP weighting at 340M. Therefore, we cannot claim that the same trend persists at larger scales.
>
> ## Q2
>
> See W2
>
> ## Q3
>
> See W7
>
> ## Q4
>
> Thank you for the question. Token order prediction cannot predict repetitive tokens sequentially. We believe this is why it performs poorly on self-speculative decoding. It still performs better because TOP objectives on predicting the order of future tokens provide lookahead signals during training, which is needed in star graph tasks.
>
> ## Q5
>
> See W4

---

> > ### Author Rebuttal · Reviewer_QZTG · 2026-04-03
> >
> > We thank the authors for their thorough response. The authors have clarified the concerns and question I had about their work. I will update the score favorably. But I still remain worried about:
> > - The low performance reported by the DS-MTP on GSM8K.
> > - Appendix A states that they take the hyper parameters of DeltaNet to train their models, yet they are using a softmax-attention transformer.

---

> > > ### Author Response · Authors · 2026-04-07
> > >
> > > We highly appreciate your considerate acknowledgment of our rebuttal, and we eagerly await the update on your score. With regard to your remaining concerns, we reply as follows:
> > >
> > > 1. We agree that the weak DS-MTP result should be treated cautiously. At present, this is the result obtained by our reproduction, and we do not have sufficient evidence to determine whether it reflects a true weakness of DS-MTP or sensitivity to implementation or training recipe. To address this concern, we will revise the paper in two ways:
> > > (1) Explicitly state that DS-MTP is a reproduction baseline under our setup, and avoid drawing strong conclusions from its underperformance.
> > > (2) Add a limitation noting that our DS-MTP reproduction may not fully reflect the best achievable performance of the method.
> > >
> > > **Our main conclusions do not depend on DS-MTP underperforming.** They are primarily supported by the comparisons against **NTP and MTP**, as well as the **star-graph results**.
> > >
> > > 2. We would like to clarify a misunderstanding here: we use established hyperparameters from the default configs of the FLA/Flame repository, which were originally used in the DeltaNet paper. That paper also includes a vanilla softmax transformer baseline trained with these hyperparameters. Since this configuration has been used in that paper and in subsequent work along this line, we found it to be a suitable baseline and setup for our paper as well.

---

### Decision · Program_Chairs · 2026-04-30

**Decision:**

Accept (regular)

**Comment:**

In this paper the authors propose token order prediction (TOP) to predict the order of the upcoming tokens in a future window in order to improve the inference performance and efficiency of LLMs. Commonly-used multi-token prediction (MTP) may underperform in some NLP applications. To deal with it, TOP is introduced as an alternative auxiliary loss based on learning-to-rank. TOP is considered an objective that is more benign to optimize compared to MTP. The authors extensively evaluate the proposed TOP using various model sizes on a variety of benchmarks including general language modeling, generative tasks, the star graph task and self-speculative decoding. They show that TOP can outperform some competing techniques such as NTP, MTP and DS-MTP. Overall all reviewers consider the work interesting which may have its value to the LLM community for improved inference performance and efficiency. The rebuttal provided by the authors has cleared most of the significant concerns raised by the reviewers. However, there are a few standing issues after the rebuttal and discussion.  1) The improvements appear to be modest.  2) The notable underperformance of DS-MTP. 3) There is a lack of theoretical justification and insights on why ranking is more helpful to learn better representations.